



# Large-Scale Vertical Velocity, Diabatic Heating and Drying Profiles Associated with Seasonal and Diurnal Variations of Convective Systems Observed in the GoAmazon2014/5 Experiment

Shuaiqi Tang[1], Shaocheng Xie[1], Yunyan Zhang[1], Minghua Zhang[2], Courtney Schumacher[3], Hannah Upton[3], Michael P. Jensen[4], Karen L. Johnson[4], Meng Wang[4], Maike Ahlgrimm[5], Zhe Feng[6], Patrick Minnis[7] and Mandana Thieman[8]

[1]Lawrence Livermore National Laboratory, Livermore, CA, 94550, USA
[2]School of Marine and Atmospheric Sciences, tony Brook University, Stony Brook, NY, 11794, USA
[3]Department of Atmospheric Sciences, Texas A&M University, College Station, TX, 77843, USA
[4]Brookhaven National Laboratory, Upton, NY, 11973, USA
[5]European Centre for Medium-Range Weather Forecasts, Shinfield Park, Reading RG2 9AX, United Kingdom
[6]Pacific Northwest National Laboratory, Richland,Washington, 99354, USA
[7]NASA Langley Research Center, Hampton, VA, 23681, USA
[8]Science Systems and Applications, Inc, Hampton, VA 23666, USA

*Correspondence to*: Shuaiqi Tang (tang32@llnl.gov)

**Abstract.** This study describes the characteristics of large-scale vertical velocity, apparent heating source ($Q_1$) and apparent moisture sink ($Q_2$) profiles associated with seasonal and diurnal variations of convective systems observed during the two intensive operational periods (IOPs) of the Green Ocean Amazon (GoAmazon2014/5) experiment, which was conducted near Manaus, Brazil in 2014 and 2015. The derived large-scale fields have large diurnal variations according to convective activity in the GoAmazon region and the morning profiles show distinct differences between the dry and wet seasons. In the wet season, propagating convective systems originating far from the GoAmazon region are often seen in the early morning, while in the dry season, they are rarely observed. Afternoon convective systems due to solar heating are frequently seen in both seasons. Accordingly, in the morning, there is strong upward motion and associated heating and drying throughout the entire troposphere in the wet season, which is limited to lower levels in the dry season. In the afternoon, both seasons exhibit weak heating and strong moistening in the boundary layer related to the vertical convergence of eddy fluxes. A set



of case studies of three typical types of convective systems occurring in Amazonia - i.e., locally-
occurring systems, coastal-occurring systems and basin-occurring systems - is also conducted to
investigate the variability of the large-scale environment with different types of convective
systems.



## 1. Introduction

Amazonia is one of the major tropical convective regions in the global climate system. It
provides moisture to the global hydrological cycle and energy to drive the global atmospheric
circulation. Understanding convective systems over the Amazon region through observations is
important for understanding and simulating global circulation and climate. However, most of
Amazonia is covered by tropical forest with only a few observational sites. In order to collect
the observations needed to improve our understanding of convective systems over Amazonia,
several major field campaigns have been conducted in this area such as the Amazon Boundary
Layer Experiments (Harriss et al., 1988; Harriss et al., 1990), the Large-Scale Biosphere-
Atmosphere Experiment in Amazonia (LBA) (Silva Dias et al., 2002b), and the CHUVA project
(Machado et al., 2014).
Recently, an internationally collaborative experiment, the Observations and Modeling of
the Green Ocean Amazon (GoAmazon2014/5) (Martin et al., 2016), was conducted in the region
around Manaus, Brazil from January 2014 to December 2015 with a focus on the aerosol and
cloud life cycles and aerosol-cloud-precipitation interactions over tropical rainforests. Two 40-
day Intensive Operational Periods (IOPs) were conducted to investigate the seasonal variations
of clouds and aerosols, as well as their interactions. IOP1 took place from 15 February to 26
March 2014 during the wet season, and IOP2 took place from 1 September to 10 October 2014
during the dry season. The goal of this study is to document and understand the seasonal
variability and diurnal cycle of large-scale vertical velocity, heat and moisture budgets associated
with the convective systems observed during the two IOPs in the GoAmazon2014/5 experiment.



55 The Amazon region has a significant seasonal variation in precipitation amount. Rainfall

56 is approximately 300 mm per month during the wet season while it is close to 100 mm per month

57 during the dry season (Tanaka et al., 2014). Many studies have examined the seasonal variation

58 of clouds and precipitation in Amazonia (e.g. Fu et al., 2001; Schumacher and Houze, 2003;

59 Machado et al., 2004; Li et al., 2006; Marengo et al., 2012). Compared to the large variation in

60 clouds and rainfall, the seasonal variation in CAPE is small (Machado et al., 2004; Martin et al.,

61 2016), which implies that small perturbations in the large-scale circulation can drive dramatic

62 changes in hydrological fields in this region. Few studies, however, have studied the seasonal

63 variation of the diabatic heating and drying structures associated with the convective systems in

64 the Amazon region.

65 The diurnal cycle of the atmosphere is an important feature that is poorly simulated in

66 climate models. Many efforts have been made to observe and to understand the diurnal cycle

67 over the Amazon basin using surface observations (e.g. Harriss et al., 1990; Cutrim et al., 2000;

68 Machado et al., 2004; Tanaka et al., 2014) or satellite data (e.g. Minnis and Harrison, 1984;

69 Greco et al., 1990; Janowiak et al., 2005; Burleyson et al., 2016). The diurnal cycle over the

70 Amazon basin is complex because it is affected by three types of convective systems: locally-

71 occurring systems (LOS) generated locally in the form of small convective cells (area less than

72 1000 km$^2$) with short life time (on the order of 1 hour), coastal-occurring systems (COS)

73 initialized at the northeast coast of Brazil by the sea-breeze and propagating inland as squall lines,

74 and basin-occurring systems (BOS) initialized in the Amazon basin in the form of mesoscale

75 convective systems (MCS) with areas larger than 1000 km$^2$ (Greco et al., 1990). These systems

76 reach Manaus, near the center of the Amazon basin, at different times of the day, causing a broad

77 peak of precipitation from morning to early afternoon (e.g. Machado et al., 2004; Tanaka et al.,



2014; Burleyson et al., 2016).  Schumacher et al. (2007) examined the diurnal cycle of the large-
scale $Q_1$ budget in the southwest Amazon during LBA, but used only two profiles per day, which
do not capture the rapidly changing environment. In addition, the diurnal cycle over the highly
deforested southwest Amazon is not necessarily representative of the more pristine central
Amazonian rainforest.

In this study we use data collected from the comprehensive GoAmazon2014/5 field

campaign to examine the seasonal and diurnal variations of the large-scale vertical velocity and
heat and moisture budgets associated with the convective systems that occur in central Amazonia.
Section 2 provides details of the data and method used to derive the large-scale profiles for the
GoAmazon2014/5 experiment.  Section 3 describes the synoptic conditions for the two IOPs.
Sections 4 and 5 show the seasonal variation and diurnal cycle of the large-scale fields,
respectively.  Section 6 further investigates three selected cases representing different types of
convective systems in the wet season.  The summary and discussion are given in Section 7.

**2.  Data and Method**

Due to the lack of an appropriate sounding array to capture the divergence and advection

fields in the analysis domain, the large-scale vertical velocity and budgets analyzed in this study
were derived by using, as a first guess, the European Centre for Medium-Range Weather
Forecasts (ECMWF) analysis data that are subsequently constrained with surface and top of
atmosphere (TOA) observations.  The upper-level fields from ECMWF analysis data are
adjusted to conserve the vertical integration of mass, moisture and dry static energy through a
constrained variational analysis technique described in Zhang and Lin (1997) and Zhang et al.





(2001). As indicated in Xie et al. (2004), the use of the surface and TOA observations as
constraints improves the quality of the large-scale vertical velocity and budgets in operational
analysis data and makes the data suitable for budget analysis and cloud modeling studies. An
important by-product of this study is the derived large-scale forcing data supporting modeling
studies, which are available to the community at the Atmospheric Radiation Measurement (ARM)
program Archive (http://iop.archive.arm.gov/arm-iop/0eval-data/xie/scm-forcing/iop_at_mao/).
Figure 1 shows the location of the GoAmazon2014/5 experiment and the analysis domain
(the red octagon, referred to as the GoAmazon domain) used in this study. The observational
research sites and major cities near the region are also shown on the map. The required surface
and TOA fluxes as the constraints for the variational analysis are constructed as follows. The
precipitation used in this study is derived from the System for the Protection of Amazonia
(SIPAM) S-band (10 cm wavelength) radar operated at Ponta Pelada airport, the center of the
GoAmazon domain. The SIPAM radar reflectivity constant altitude plan position indicator
(CAPPI) at 2.5 km above ground was used to generate the rain rate products using a single Z-R
relation of $Z = 174.8 R^{1.56}$ derived from Joss-Waldvogel disdrometer data obtained by the
CHUVA campaign near Manacapuru during the wet season of early 2014. Other surface
constraint variables, such as surface radiative fluxes and latent and sensible heat fluxes, are
obtained from the broadband radiometer (ARM Climate Research Facility, 1994) and eddy
correlation flux measurement system (ARM Climate Research Facility, 2003) at the ARM
Mobile Facility site near Manacapuru (3.213°S, 60.598°W; "ARM site" in Figure 1).
Observations of latent and sensible heat fluxes at two other Brazilian research sites - K34
("FLUXNET-BR Ma2" in Figure 1) and the Amazon Tall Tower Observatory ("ATTO Tower"
in Figure 1) - are also used. Because of the limited number of surface sites, it is challenging to



obtain domain mean fluxes that can well represent the analysis domain. In this study, we use the
Cressman's objective analysis method (Cressman, 1959) to incorporate these limited
observations into the analysis with the ECMWF analysis as the first guess. The TOA
measurements of broadband radiative fluxes are estimated from the Thirteenth Geostationary
Operational Environmental Satellite (GOES-13) 4-km visible (0.65 μm) and infrared window
(10.8 μm) radiances using the narrowband-to-broadband (NB-BB) conversion method of Minnis
and Smith (1998) that was updated similar to Khaiyer et al. (2010), with some modifications to
more closely match those measured by the Clouds and Earth's Radiant Energy System (CERES)
on the Aqua and Terra satellite. All data are interpolated into 3 h and 25 hPa (if applicable)
temporal and vertical resolutions, respectively.

**3.  Background of Synoptic Conditions**

The IOP-averaged sea-level pressure and 10-meter horizontal winds from ERA-Interim

reanalysis (Dee et al., 2011) are plotted in Figure 2. During IOP1, the Atlantic Intertropical
Convergence Zone (ITCZ) was located near the Equator; while during IOP2, it was located near
10°N. A fourteen-day trajectory study shows that the air masses over Manaus typically come
from the Northern Hemisphere during IOP1 and from the Southern Hemisphere during IOP2
(Martin et al., 2016). The top three rows of Figure 3 show the domain-averaged zonal (u) wind,
meridional (v) wind, and relative humidity relative to liquid water, from the adjusted ECMWF
analysis. Consistent with those derived from radiosonde data in Martin et al. (2016), IOP1 was
dominated by northeasterly winds in the lower troposphere, with moist air throughout the



troposphere; IOP2 was dominated by easterly winds in the lower troposphere, with a dry free
troposphere.

The cloud frequency and domain-mean precipitation observed during IOP1 and IOP2 are

shown in the remaining two rows of Figure 3. The cloud frequency was derived from the Active
Remote Sensing of Clouds (ARSCL) (Kollias et al., 2007) product, which uses a combination of
the 95GHz W-band ARM cloud radar (WACR), micropulse lidar (MPL), and ceilometer located
at the ARM site to determine a best-estimate cloud mask with 5-second temporal and 30-meter
vertical resolution. The ARSCL product leverages each instrument's strengths: the WACR
penetrates non-precipitating thick clouds, the MPL is sensitive to thin clouds, and the ceilometer
reliably detects cloud base. The ARSCL-derived cloud mask data were then used to produce 3-
hourly cloud frequencies following the method described in Xie et al. (2010b). The wet season
has more cloud and precipitation events than the dry season. However, the convective systems
in the dry season are typically more intense than those occurring in the wet season (Giangrande
et al., 2016, accepted). Compared to 15-year climatology, the precipitation around Manaus
during 2014 has a positive anomaly in IOP1 and negative anomaly in IOP2 (Burleyson et al.,
2016; Martin et al., 2016). Nevertheless, the annual cycle in 2014 is still broadly representative
of the climatology (Burleyson et al., 2016).

**4. Seasonal Variation**

In this section, we focus on the contrast between the dry and wet season large-scale

vertical velocity and energy and moisture budgets. The upper row of Figure 4 shows the
temporal evolution of large-scale vertical velocity in IOP1 (wet season, left) and IOP2 (dry



season, right), and the IOP-mean profiles are shown as the black solid lines in the bottom row.
We also define rainy (black dotted lines) and non-rain periods (gray lines) using a threshold of
0.2 mm hr$^{-1}$. A value of 0.2 mm hr$^{-1}$ rather than 0 mm hr$^{-1}$ is used because in some cases ground
clutter in the SIPAM radar data may be misinterpreted as light precipitation. Changing the
threshold affects the magnitude of the vertical profiles but does not change the seasonal contrast
and the results of this study. Using this threshold, the percentage of the rainy period to the entire
IOP is 36.9% for IOP1, but is 17.8% for IOP2, indicating that the rain frequency is an important
factor impacting the seasonal mean contrast. The red and blue lines represent the mean profiles
of morning (at 5 local time (LT)) precipitation systems and afternoon (at 14 LT) precipitation
systems, respectively, which will be discussed in Section 5.
The non-rain vertical velocity profiles are relatively weak, with downward motion
dominating in the upper troposphere during both dry and wet seasons. The rainy vertical
velocity profiles show strong upward motion throughout the troposphere during both IOPs, but
the level of maximum upward motion is different. The upward motion during the rainy period of
IOP1 has a broad peak structure from ~700 to 300 hPa with the maximum at ~350 hPa. The
350-hPa upward motion peak is consistent with that shown in the Tropical Ocean and Global
Atmosphere Coupled Ocean-Atmosphere Response Experiment (TOGA COARE) (Lin and
Johnson, 1996), but lower than the peak of ~265 hPa observed in the Tropical Warm Pool-
International Cloud Experiment (TWP-ICE) (Xie et al., 2010a). The upward motion during the
IOP2 rainy period also has a broad peak but the maximum is at a much lower level (~550 hPa)
than in IOP1. Because the frequency of the rainy period is higher in IOP1 than in IOP2, the IOP-
mean upward motion is stronger during IOP1 but weaker and limited to the lower troposphere
during IOP2. As discussed in the next section, the difference in morning precipitation systems





largely contributes to the seasonal contrast in the vertical velocity profiles between the wet and
dry seasons.

Figures 5 and 6 show the temporal evolution and IOP-mean of apparent heating $Q_1$ and

apparent drying $Q_2$ profiles, respectively.  $Q_1$ and $Q_2$ were first introduced by Yanai et al. (1973)
to estimate the diabatic processes:
$$Q_1 = \frac{\partial \bar{s}}{\partial t} + \bar{\bar{V}} \cdot \nabla \bar{s} + \bar{\omega} \frac{\partial \bar{s}}{\partial p}$$
$$= Q_{rad} + L_v (c - e) - \frac{\partial \overline{\omega' s'}}{\partial p} \quad ,$$
(1)

$$Q_2 = -L_v \left( \frac{\partial \bar{q}}{\partial t} + \bar{\bar{V}} \cdot \nabla \bar{q} + \bar{\omega} \frac{\partial \bar{q}}{\partial p} \right)$$
$$= L_v (c - e) + L_v \frac{\partial \overline{\omega' q'}}{\partial p} \quad ,$$
(2)

where  $s = C_p T + gz$ is the dry static energy;  $q$ is water vapor mixing ratio;  $\vec{V}$ is horizontal wind
vector;  $\omega$ is vertical velocity in pressure coordinate;  $Q_{rad}$ is radiative heating;  $L_v (c - e)$ is the
latent heat from water condensation and evaporation (in general it also includes the latent heat
and water vapor change from ice phase change);  the overbar refers to a horizontal average and
the prime refers to a deviation from the average. $Q_1$ and $Q_2$ are calculated from the large-scale
dynamics (the first lines of the equations) and represent the unresolved physical heat sources and
moisture sinks (the second lines).  The vertical distributions of heating and drying profiles are
important to the large-scale circulation as discussed in many other studies (e.g. Hartmann et al.,
1984; Lau and Peng, 1987; Puri, 1987; Hack and Schubert, 1990).





Similar to the profiles of vertical velocity, non-rain $Q_1$ and $Q_2$ profile magnitudes in both
IOPs are weak with small amounts of heating and moistening below 600 hPa indicative of non-
precipitating or very weakly precipitating shallow cumulus and congestus clouds (Schumacher et
al., 2008).  Rainy period $Q_1$ and $Q_2$ profiles show strong heating and drying throughout the
troposphere during both IOPs associated with deep convection, and both of them have double
peak structures that vary between dry and wet seasons.  $Q_1$ during IOP1 has a broad primary
peak between 600 and 400 hPa, while the primary $Q_1$ peak during IOP2 maximizes more sharply
at 550 hPa.  The secondary peaks of $Q_1$ are at ~750 hPa in both IOPs.  The peaks of $Q_2$ in IOP1
(at 500 and 750 hPa) are higher than those in IOP2 (at 650 and 800 hPa).  The double peak
features of $Q_1$ and $Q_2$ are likely due to different physical processes.  For $Q_1$, the local minimum
usually occurs near the melting level (~600 hPa), indicating latent cooling due to ice melting.
Because the melting level is nearly constant in the tropics, the local minimums of $Q_1$ are more or
less at the same level as seen in other tropical field campaigns (e.g. Schumacher et al., 2008; Xie
et al., 2010a).  For $Q_2$, the double-peak structure is the combined effect of convective (lower
peak) and stratiform (higher peak) rain production (Lin and Johnson, 1996).  The peak levels for
stratiform and convective clouds may vary in different locations and times such as in the two
IOPs in this study.

**5.  Diurnal Cycle**

The diurnal cycles of domain mean radar-derived precipitation and surface CAPE and

convective inhibition (CIN) for both IOPs are plotted in Figure 7.  Precipitation in IOP1 extends
from early morning to afternoon, consistent with Tanaka et al. (2014).  In IOP2, most of the





precipitation occurs in the afternoon.  The magnitude of afternoon precipitation in IOP2 is just
slightly smaller than that in IOP1, but the magnitude of morning precipitation in IOP2 is
significantly lower than that in IOP1, indicating that the differences between dry and wet seasons
are mainly due to the morning precipitation events.  The surface CAPE has similar magnitudes in
the daytime during IOP1 and IOP2, but in the early morning it rises later and slower during IOP1
than during IOP2, probably because early morning precipitation during IOP1 has released
atmospheric instability.  The surface CIN is typically small, especially during IOP1, which is due
to the high surface relative humidity over the Amazon rainforest.
The diurnal cycles of cloud frequency, large-scale vertical velocity, $Q_1$, $Q_2$ and $Q_1 - Q_2$
for IOP1 (left) and IOP2 (right) are shown in Figure 8.  Derived from Eq. (1) and (2),
$$Q_1 - Q_2 = Q_{rad} - \frac{\partial \overline{\omega' h'}}{\partial p} \qquad (3)$$
where  $h = s + L_v q$  is the moist static energy.  With the phase change of water vapor cancelled,
$Q_1 - Q_2$  represents the radiative effect and the vertical convergence of eddy fluxes.
Consistent with the diurnal cycles of precipitation, the observed clouds and large-scale
vertical velocity differ primarily in the morning between IOP1 and IOP2.  In IOP1, the early
morning upward motion peaks at 700 hPa and extends to the upper troposphere around 200 hPa.
The early afternoon upward motion peaks at the upper troposphere and extends above 100 hPa.
Accordingly, clouds are mainly seen between 800 and 500 hPa in the early morning but
throughout the entire troposphere in the afternoon.  In IOP2, morning convective systems are
generally limited to the lower levels, as shown by weak upward motion below 600 hPa and
downward motion above.  Thus, few clouds are observed in the lower and middle troposphere





while some high clouds remain from the previous day's convective activities. The afternoon
convective systems are strong and deep in both IOPs, with upward motion in the upper
troposphere associated with convective cloud growth and downward motion in the lower
troposphere associated with convective downdrafts.

Consistent with the clouds and vertical velocity, Figure 8 also shows significant seasonal

differences of $Q_1$ and $Q_2$ profiles in the morning, with heating and drying extending to the upper
troposphere in IOP1 but cooling and moistening above 600-650 hPa in IOP2. In the afternoon,
both IOPs show strong heating and drying in the middle and upper troposphere with weak
heating and strong moistening occurring below 700 hPa. The low-level heating and moistening
feature has been observed in trade wind regimes during westerly wind bursts and monsoon break
periods (Nitta and Esbensen, 1974; Lin and Johnson, 1996; Johnson and Lin, 1997; Xie et al.,
2010a), in which the vertical convergence of eddy fluxes and detrainment of shallow cumulus
were considered as the causes. In this study it is also seen in the afternoon precipitating periods
(red lines in Figure 5 and 6). The last row in Figure 8 shows $Q_1 - Q_2$ where two positive centers
are seen during daytime at ~750 to 950 hPa and ~250 to 550 hPa, respectively. Considering the
two terms in the right-hand-side of Eq. (3), the troposphere usually has a radiative cooling effect
and therefore $Q_{rad}$ is usually negative. The positive $Q_1 - Q_2$ has to be due to the vertical
convergence of eddy fluxes of moist static energy associated with convective process, where
positive $Q_1$ comes from vertical convergence of dry static energy flux, and negative $Q_2$ comes
from vertical convergence of moisture flux.

**6. Case Studies**



A set of case studies is conducted to further understand the large-scale vertical velocity
and heat and moisture budgets for the three typical types of convective systems that often occur
in the wet season in Amazonia: locally-occurring systems (LOS), coastal-occurring systems
(COS), and basin-occurring systems (BOS).  Previous studies have found that LOS often occur
in the afternoon characterized as scattered convections generated through solar heating at the
surface, while most COS and BOS are propagating systems associated with mid-level easterlies
and westerlies, respectively (e.g. Cifelli et al., 2002; Silva Dias et al., 2002a; Williams et al.,
2002), and affect Manaus in the early morning.  COS occurring in easterlies are often westward
propagating squall-lines with intense leading lines that are more vertically developed.  BOS
generated in the westerlies are generally less vertically developed MCSs with a broad horizontal
area and relatively homogeneous precipitation extending over a long time (Cifelli et al., 2002).
Table 1 gives the number of each type of precipitation system observed during the two IOPs,
identified from the radar loop (available at
https://www.youtube.com/playlist?list=PLVqbwaasmlvtcu2kl_U5RaaNF0kYqW6ua) and the satellite
infrared images (available at http://www-pm.larc.nasa.gov/).  The two BOS cases identified in
IOP2 both occurred in the Amazon basin, but their structures are more like COS as squall lines
propagating westward.  There are more COS and BOS in IOP1 than in IOP2, but the number of
afternoon LOS in IOP1 is just slightly higher than that in IOP2.  This again indicates that the
frequency of morning propagating convective systems contributes to the variation of the diurnal
cycle between the wet and dry seasons.
The three selected cases are a LOS starting from 11 LT, 13 March 2014, a COS starting
from 2 LT, 20 March 2014 and a BOS starting from 17 LT, 1 March 2014.  The times of these
events are marked by the black lines in Figure 3.  Mid-level wind was dominated by westerlies



on 1 March (day 60) and easterlies on 20 March (day 79).  Figure 9 shows representative scans
of the radar reflectivity at elevation angle of 0.9° for these three cases, as well as the time series
of the domain mean precipitation.  The LOS case has many small-scale scattered convective cells
that last for very short times (typically a couple of hours).  Because of the small horizontal
coverage of the convective cells, the domain mean precipitation is less than that in the other two
cases.  The COS case has a clear bow-shape echo indicating a squall line front.  The horizontal
size of the precipitating system is about 100 km and it moves quickly westward.  The BOS case
has a much larger horizontal area of moderate precipitation with some embedded convective
cells.  It moves southeastward and lasts more than 10 hours over the GoAmazon domain.

The point-observed cloud frequency and domain-averaged surface CAPE and CIN, u-

and v-winds, relative humidity, large-scale vertical velocity, $Q_1$, and $Q_2$ for the three cases are
shown in Figures 10-12, respectively.  For the LOS case, the cloud frequency is much smaller
than in the other two cases, since the convective cells have small horizontal extent and only
occupy a small portion of the region.  A shallow-to-deep transition of convective clouds can be
seen.  The surface CAPE is large, with weak mid-level winds and moist air at the surface before
the convection occurred.  Upward motion corresponds to the deep convection, and the magnitude
is smaller than in the other two cases, consistent with weaker precipitation.  Starting around 9 LT,
$Q_1$ shows diabatic heating throughout the troposphere during the deep convection, while $Q_2$
shows strong moistening between 750 and 950 hPa and weak drying above that layer.  The
daytime moistening between 750 and 950 hPa due to the vertical convergence of eddy moisture
fluxes can also be seen on many other days during the two IOPs and was discussed in Section 5.
Note that there is a time lag between observed cloud frequency and the domain-averaged large-
scale fields, which might be partially due to the fact that the cloud frequency observations were



taken from vertically pointing instruments at the ARM site 67.8 km downwind of the center of
the GoAmazon domain.

The COS (Figure 11) and BOS (Figure 12) cases both show a shallow-to-deep convective

cloud transition from the previous evening to late afternoon, with a moist lower-level atmosphere.
Both cases have smaller surface CAPE than the LOS case, possibly because the convective
systems have released the atmospheric instability in the morning.  The COS case passed through
the GoAmazon domain between 6 and 12 LT in strong mid-level easterlies, with deep clouds and
strong upward motion associated with the squall line.  Stratiform clouds, associated with weak
upward motion, remained in the upper levels until ~16 LT.  Condensation from the deep
convection contributes to strong diabatic heating and drying throughout the troposphere, while
after the passage of the squall line, a few isolated convective cells moved in and the large-scale
structure becomes similar to that in the LOS case, with upper-level heating and drying, low-level
heating and moistening.  The BOS case entered the GoAmazon domain earlier than the COS case.
In weak mid-level westerlies and descending mid-to-low-level northerlies, the system moved
slowly southeastward and remained in the domain for a longer time.  Strong upward motion
related to the MCS is seen from 18 to 6 LT.  Large diabatic heating and drying related to the
strong condensation is also seen.  The remnant high clouds were maintained until ~18 LT with
precipitation weakening over time.  The upper-level heating and drying, lower-level cooling and
moistening indicate that there are precipitating stratiform clouds in the upper level and
evaporation of precipitation underneath.

**7.  Summary and Discussion**



This study presented the characteristics of the seasonal variation and diurnal cycle of the
large-scale vertical velocity and diabatic heating ($Q_1$) and drying ($Q_2$) profiles for the two IOPs
conducted during the GoAmazon2014/5 experiment.  A constrained variational analysis method
was used to derive the large-scale vertical velocity and $Q_1$ and $Q_2$ profiles based on surface and
TOA observations and ECMWF analysis.  The derived profiles correspond well with observed
clouds and precipitation describing convective systems over Amazonia.
The large-scale environment over the region near Manaus has distinct seasonal variations
and diurnal cycles.  The wet season (IOP1) has more frequent precipitation events than the dry
season (IOP2), especially in the morning.  The large-scale upward motions during rainy periods
have similar strength in both IOPs, however, the peak level in IOP1 is much higher than that
exhibited in IOP2 (350 hPa vs. 550 hPa).  $Q_1$ and $Q_2$ both have a double-peak feature during
rainy period, but the physical mechanism may be different: the double peak of $Q_1$ may be due to
the cooling near the melting level while the double peak of $Q_2$ may be due to the different height
of convective and stratiform systems.  The seasonal contrast is mainly due to the higher
occurrence of morning mesoscale convective systems observed during IOP1.  In the morning,
upward motion peaks at ~700 hPa and extends to the upper troposphere during IOP1, while it is
limited to the lower levels with downward motion at the upper levels during IOP2.  Afternoon
convective systems have a higher vertical motion peak than their morning counterparts, and both
IOPs show similar vertical structures for the afternoon systems. The large-scale vertical velocity
shows upward motion above 700 hPa and downward motion below.  Accordingly, $Q_1$ and $Q_2$
also exhibit middle and upper level heating and drying related to the deep convection.  Below
750 hPa, the profiles show relatively weak heating and strong moistening.  This heating and



moistening feature is due to the vertical convergence of eddy fluxes of heat and moisture in the
boundary layer.
Three cases from IOP1 representing different types of convective systems that often
occur in the region were chosen and analyzed in this study: locally-occurring systems (LOS),
coastal-occurring systems (COS) and basin-occurring systems (BOS). The LOS case was
characterized by many scattered and short-lived convective cells. It had relatively weak upward
motion, heating and drying in the free troposphere, and heating and moistening in the boundary
layer. The COS case occurred in strong mid-level easterlies. It was characterized as a squall line
with deep strong profiles of upward motion, heating and drying. The BOS case mainly happened
in weak mid-level westerlies and descending mid-to-low-level northerlies. It was characterized
as widespread, moderate precipitation with embedded convective cells, and lasted much longer
than the other two systems. The precipitating stratiform clouds remained at upper levels for
several hours evident by upper-level condensational heating and lower-level evaporative cooling.
The frequency of LOS cases is similar in both IOPs while the COS and BOS events occur much
more often during the wet season than the dry season. The seasonal variation of the diurnal cycle
of precipitation, clouds, and environmental variables is mainly due to the COS and BOS events
observed in the morning.
Previous studies have also shown that the river breeze has an important influence on the
diurnal cycle near the Amazon River (e.g. dos Santos et al., 2014; Tanaka et al., 2014; Burleyson
et al., 2016) and that the impact of the local circulation can extend as far as 50 km away from the
river. This local circulation and the horizontal inhomogeneity of large-scale vertical velocity,
heating, and moistening could be better studied using high-resolution 3-D gridded large-scale



forcing data from the three-dimensional constrained variational analysis recently developed by
Tang and Zhang (2015) and Tang et al. (2016). This will be the subject of a future study.

*Acknowledgment: The authors gratefully thank Luiz Machado, Jiwen Fan and many others in*
*the GoAmazon group for valuable discussions about the synoptic and climate features in*
*Amazonia region. This research is supported by the Biological and Environmental Research*
*Division in the Office of Sciences of the US Department of Energy (DOE). Work at LLNL was*
*supported by the DOE Atmospheric Radiation Measurement (ARM) program and performed*
*under the auspices of the U. S. Department of Energy by Lawrence Livermore National*
*Laboratory under contract No. DE-AC52-07NA27344. Work at Stony Brook was supported by*
*the Office of Science of the U. S. Department of Energy and by the National Science Foundation.*
*This manuscript has been authored by employees of Brookhaven Science Associates, LLC with*
*support from the ARM program and Atmospheric Systems Research Program under Contract No.*
*DE-AC02-98CH10886 with the U.S. Department of Energy. Dr. Zhe Feng at the Pacific*
*Northwest National Laboratory (PNNL) is supported by the U.S. DOE, as part of the*
*Atmospheric System Research (ASR) Program. PNNL is operated for DOE by Battelle Memorial*
*Institute under contract DE-AC05-76RL01830. The satellite analyses are supported by the DOE*
*ARM and ASR Program sunder contract, DE-SC0013896. The publisher by accepting the*
*manuscript for publication acknowledges that the United States Government retains a non-*
*exclusive, paid-up, irrevocable, world-wide license to publish or reproduce the published form of*
*this manuscript, or allow others to do so, for United States Government purposes. We thank The*
*Brazilian National Institute of Amazonian Research (INPA), the Amazonas State University*
*(UEA) and Dr. Antonio Manzi for providing surface flux data.*



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





**Figure and Table Captions:**

Table 1: number of convective systems identified in the morning and afternoon during IOP1 and IOP2.

Figure 1: The location of GoAmazon site (top) and the analysis domain for this study (bottom). The SIPAM radar is located at Ponta Pelada (indicated by red pentagram). Locations of other cities and measurement sites are also indicated.

Figure 2: The sea-level pressure (shaded) and 10-meter horizontal wind (vector) averaged for IOP1 (left) and IOP2 (right). The pentagram indicates the location of GoAmazon site.

Figure 3: Domain averaged time series of (from top to bottom) horizontal (u) wind, meridional (v) wind, relative humidity, cloud frequency (point observation at the ARM site) and precipitation for IOP1 (left) and IOP2 (right). The blank areas in cloud frequency indicate missing data. The three straight black lines in IOP1 show the three cases chosen in section 6.

Figure 4: The time series (top) and temporal mean profiles (bottom) of large-scale vertical velocity for IOP1 (left) and IOP2 (right).

Figure 5: The time series (top) and temporal mean profiles (bottom) of apparent heating source $Q_1$ for IOP1 (left) and IOP2 (right).

Figure 6: The time series (top) and temporal mean profiles (bottom) of apparent moisture sink $Q_2$ for IOP1 (left) and IOP2 (right).

Figure 7: The diurnal cycle of precipitation (up) and CAPE and CIN (bottom) for both IOPs.

Figure 8: The diurnal cycle of (from top to bottom) cloud frequency, large-scale vertical velocity, $Q_1$, $Q_2$ and $Q_1 - Q_2$ for IOP1 and IOP2. The black lines are zero-lines.

Figure 9: SIPAM radar reflectivity snapshots (left) and time series of domain-mean precipitation (right) for three cases of precipitating systems. From top to bottom: LOS, COS and BOS. The black octagons indicate the GoAmazon domain, and the red arrows indicate the propagating direction of the system.

Figure 10: The time series of (a) cloud frequency, (b) surface CAPE and CIN, (c) u wind, (d) v wind, (e) relative humidity, (f) vertical velocity, (g) $Q_1$ and (h) $Q_2$ for the LOS case. The black lines are zero-lines. The shaded and white areas in (b) indicate nightime and daytime.

Figure 11: The time series of (a) cloud frequency, (b) surface CAPE and CIN, (c) u wind, (d) v wind, (e) relative humidity, (f) vertical velocity, (g) $Q_1$ and (h) $Q_2$ for the COS case. The black lines are zero-lines. The shaded and white areas in (b) indicate nightime and daytime.

Figure 12: The time series of (a) cloud frequency, (b) surface CAPE and CIN, (c) u wind, (d) v wind, (e) relative humidity, (f) vertical velocity, (g) $Q_1$ and (h) $Q_2$ for the BOS case. The black lines are zero-lines. The shaded and white areas in (b) indicate nightime and daytime.





| | IOP1 | | IOP2 | |
|---|---|---|---|---|
| | Morning | Afternoon | Morning | Afternoon |
| Locally Occurring Systems (LOS) | 0 | 19 | 0 | 14 |
| Coastal Occurring Systems (COS) | 7 | 6 | 0 | 3 |
| Basin Occurring Systems (BOS) | 3 | 3* | 2** | 0 |

* the afternoon BOS are continued from the morning time.
** the two BOS in IOP2 are initialized in the Amazon basin but propagating westward as squall
lines.

Table 1: number of convective systems identified in the morning and afternoon during IOP1 and
IOP2.





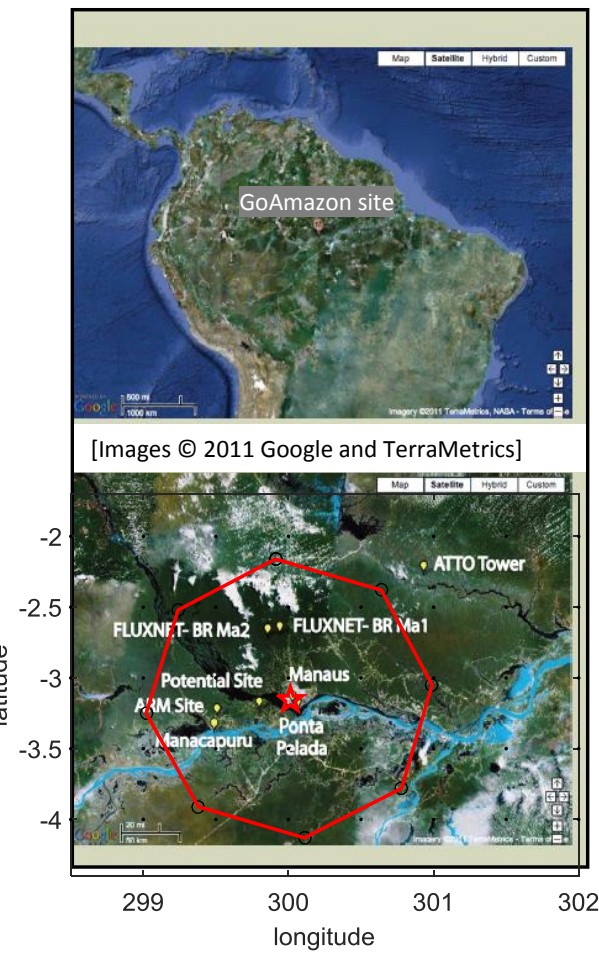




Figure 1: The location of GoAmazon site (top) and the analysis domain for this study (bottom). The
SIPAM radar is located at Ponta Pelada (indicated by red pentagram). Locations of other cities and
measurement sites are also indicated.










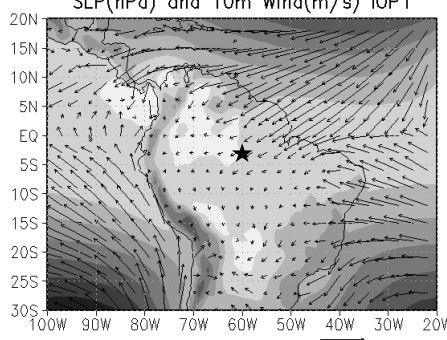 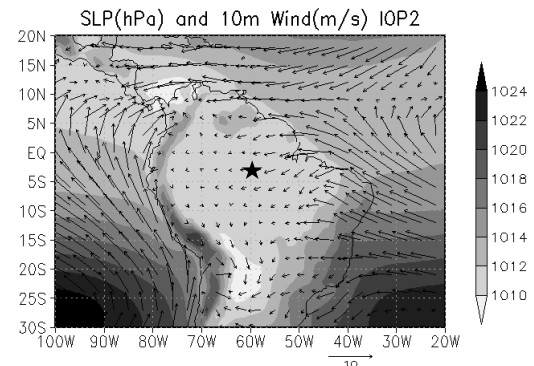


Figure 2: The sea-level pressure (shaded) and 10-meter horizontal wind (vector) averaged for IOP1 (left) and IOP2 (right). The pentagram indicates the location of GoAmazon site.













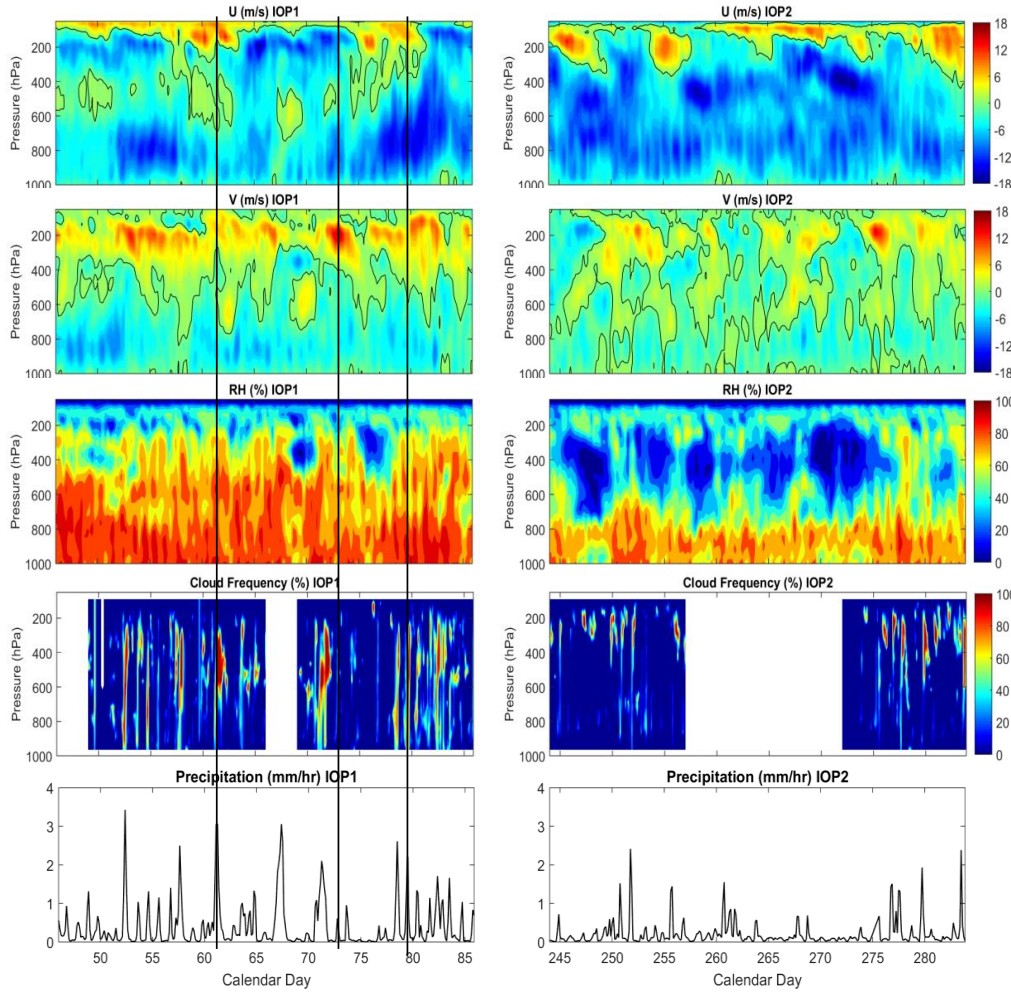


Figure 3: Domain averaged time series of (from top to bottom) horizontal (u) wind, meridional (v) wind,
relative humidity, cloud frequency (point observation at the ARM site) and precipitation for IOP1 (left)
and IOP2 (right). The blank areas in cloud frequency indicate missing data. The three straight black lines
in IOP1 show the three cases chosen in section 6.






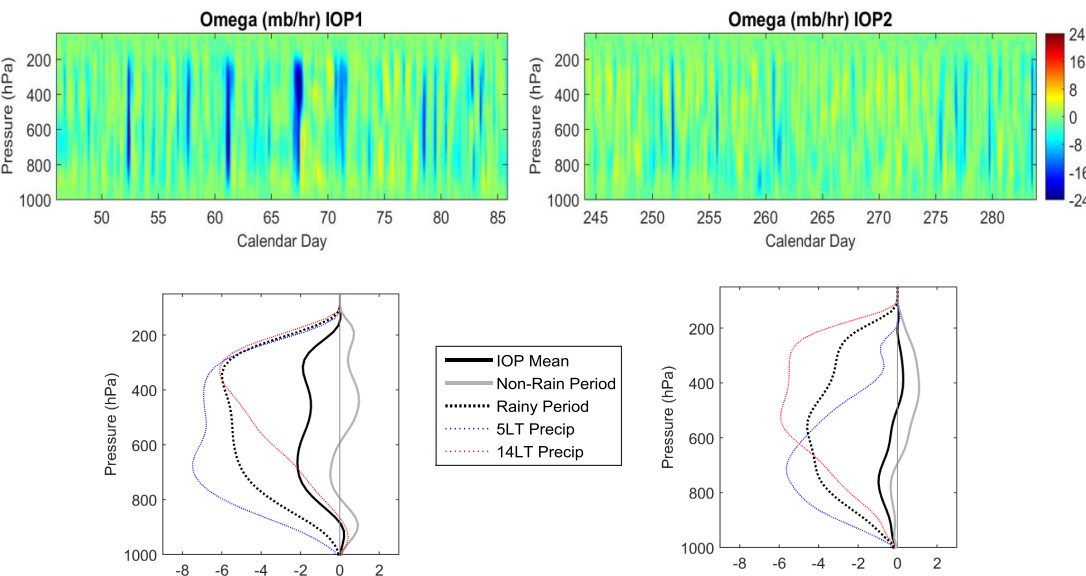




Figure 4: The time series (top) and temporal mean profiles (bottom) of large-scale vertical velocity for
IOP1 (left) and IOP2 (right).













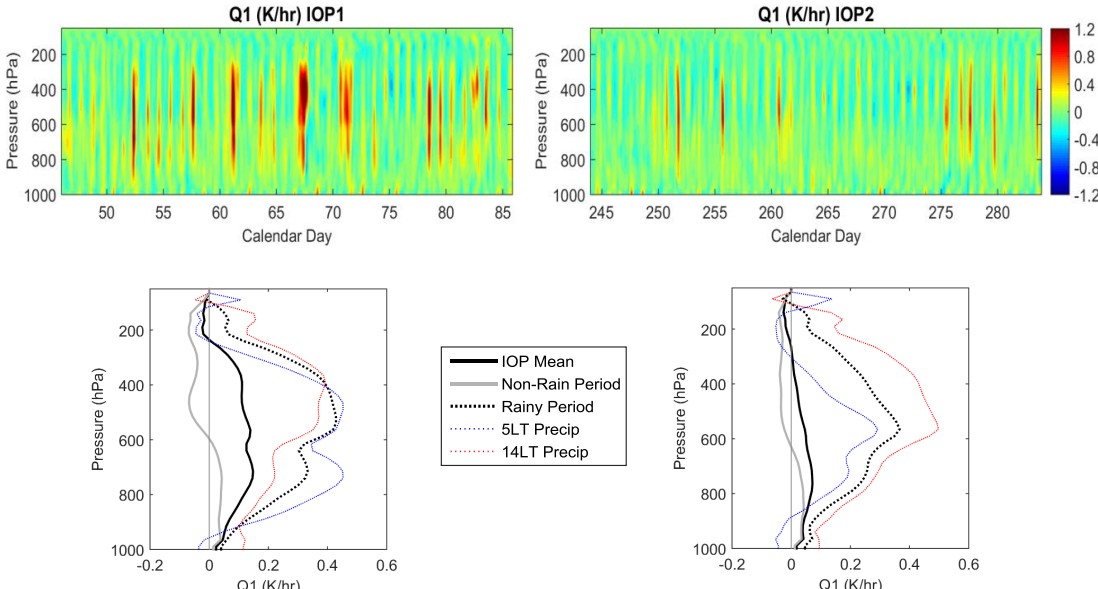




Figure 5: The time series (top) and temporal mean profiles (bottom) of apparent heating source $Q_1$ for
IOP1 (left) and IOP2 (right).














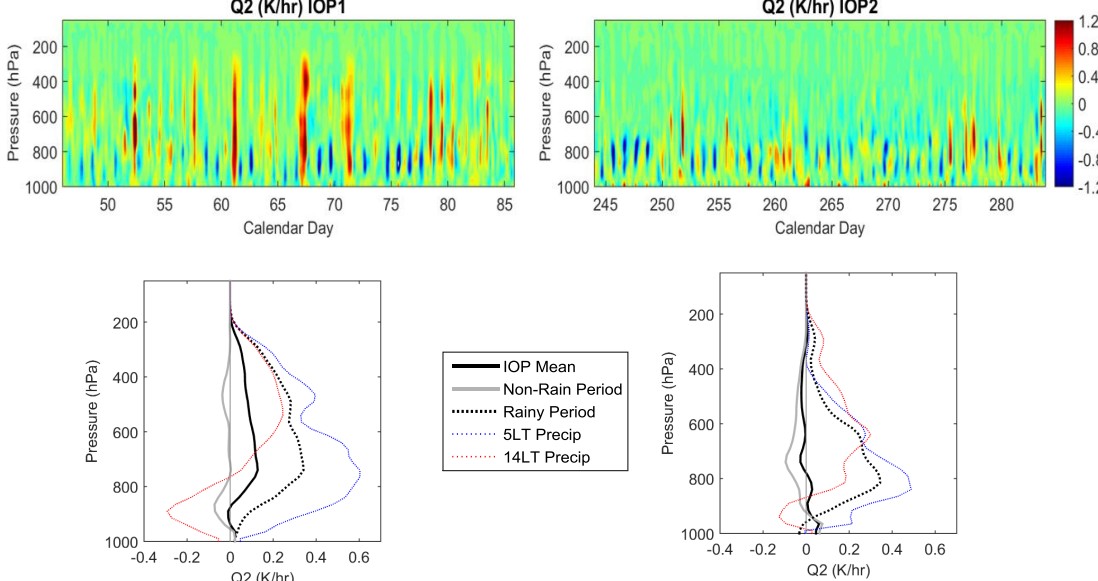


Figure 6: The time series (top) and temporal mean profiles (bottom) of apparent moisture sink $Q_2$ for
IOP1 (left) and IOP2 (right).
















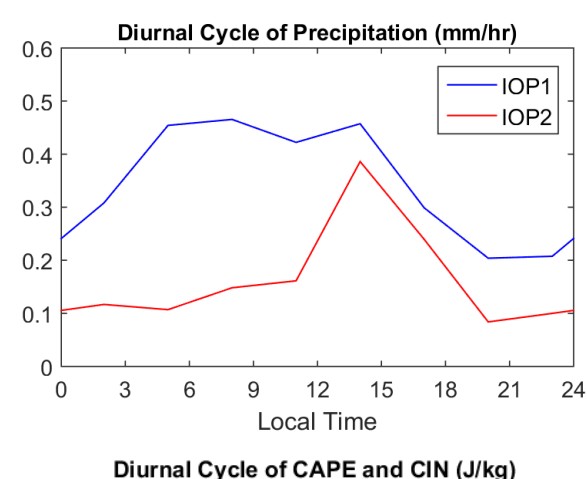

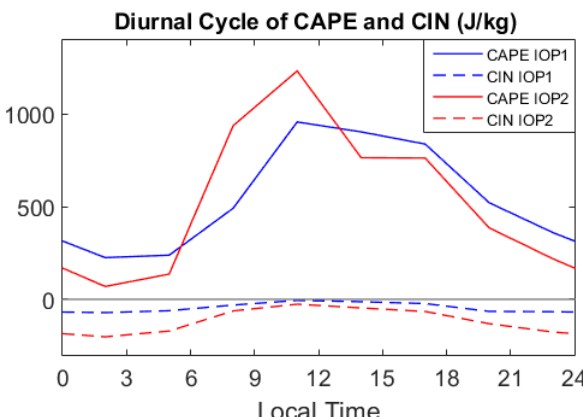


Figure 7: The diurnal cycle of precipitation (up) and CAPE and CIN (bottom) for both IOPs.








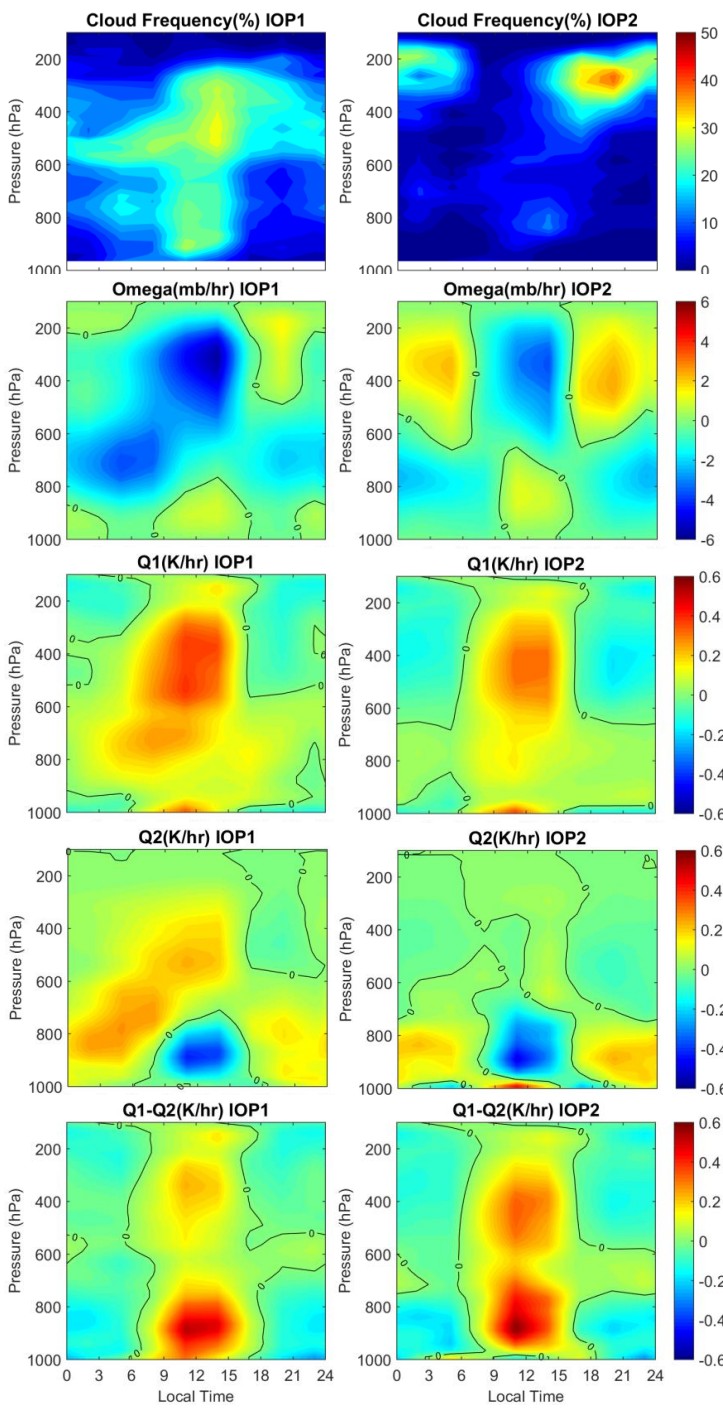


Figure 8: The diurnal cycle of (from top to bottom) cloud frequency, large-scale vertical velocity, $Q_1$, $Q_2$
and $Q_1 - Q_2$ for IOP1 and IOP2. The black lines are zero-lines.








Figure 9: SIPAM radar reflectivity snapshots (left) and time series of domain-mean precipitation (right)
for three cases of precipitating systems. From top to bottom: LOS, COS and BOS. The black octagons
indicate the GoAmazon domain, and the red arrows indicate the propagating direction of the system.






## LOS (14 March 2014)

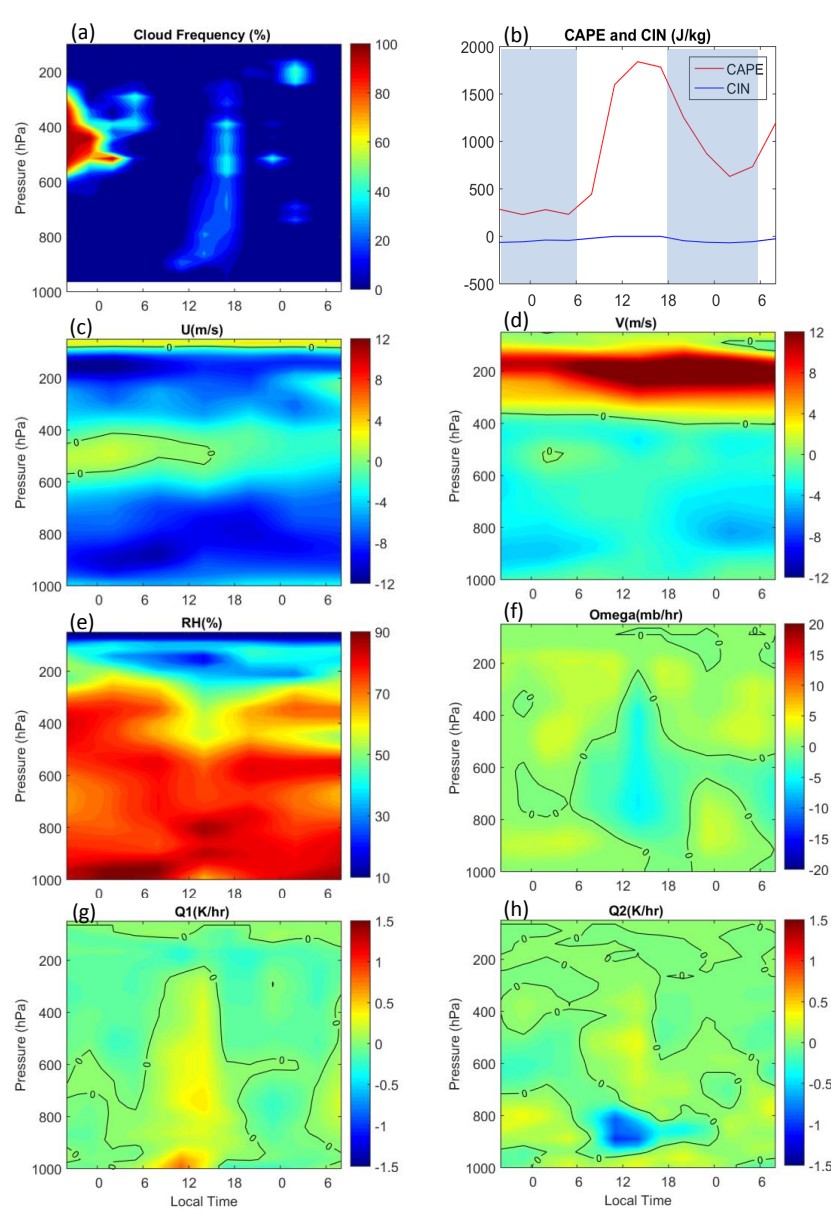


Figure 10: The time series of (a) cloud frequency, (b) surface CAPE and CIN, (c) u wind, (d) v wind, (e)
relative humidity, (f) vertical velocity, (g) $Q_1$ and (h) $Q_2$ for the LOS case. The black lines are zero-lines.
The shaded and white areas in (b) indicate nightime and daytime.




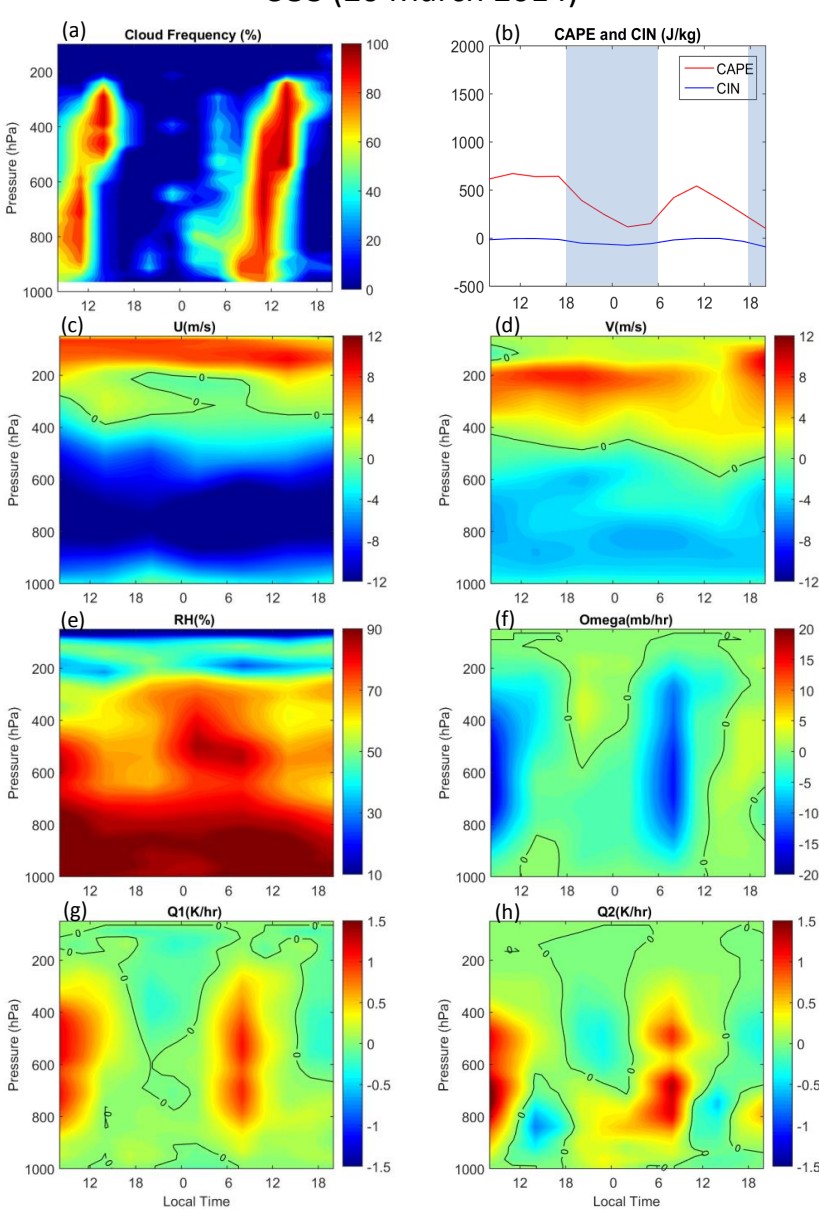


Figure 11: The time series of (a) cloud frequency, (b) surface CAPE and CIN, (c) u wind, (d) v wind, (e)
relative humidity, (f) vertical velocity, (g) $Q_1$ and (h) $Q_2$ for the COS case. The black lines are zero-lines.
The shaded and white areas in (b) indicate nightime and daytime.




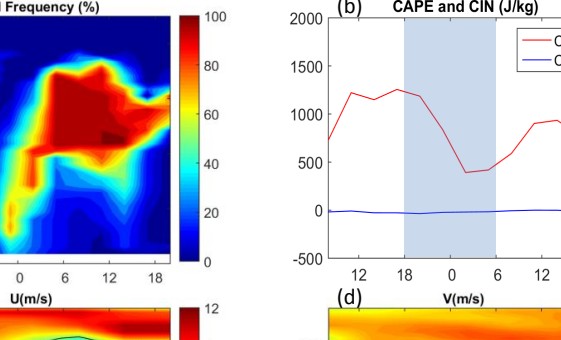

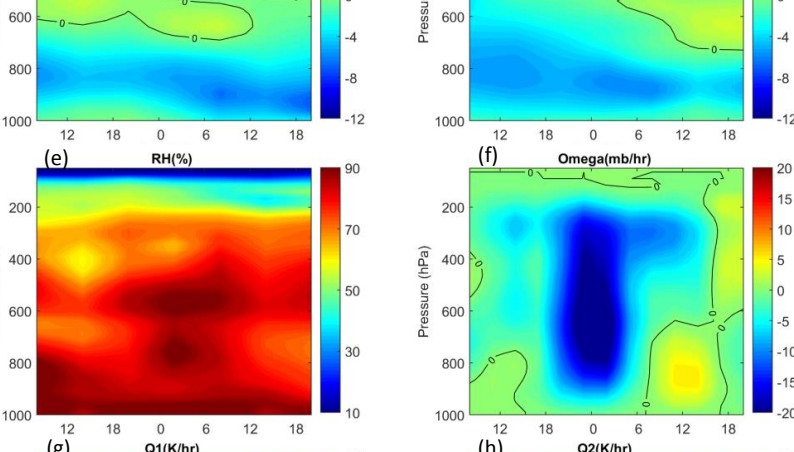


Figure 12: The time series of (a) cloud frequency, (b) surface CAPE and CIN, (c) u wind, (d) v wind, (e)
relative humidity, (f) vertical velocity, (g) $Q_1$ and (h) $Q_2$ for the BOS case. The black lines are zero-lines.
The shaded and white areas in (b) indicate nightime and daytime.