# Peer review of "Large-Scale Vertical Velocity, Diabatic Heating and Drying"

_Atmospheric Chemistry and Physics, 2016_

## Referee Comment (RC1) · A. K. Betts (Referee) · 2 Aug 2016

This is an interesting discussion of two IOPs during Go-Amazon, one in the wet season and the other the dry season; with three cases studies from the wet season of local, coastal and basin scale convection. It is suitable for publication with a little revision. It is written for a 'club' and needs clarity for less specialized readers.

You mention 2014-2015, but the only data you show is from 2014. Figure 1 has a

'potential site'? L93 - see below.

P10. Although equations (1) and (2) are based on historic literature, the units are not well defined here, and Q1, Q2 do not actually have the units of K/hr as in Figure 8. Nor does QR in (1).

I did not find the Q1-Q2 discussion very satisfactory for Figure 8. Do you have an estimate for QR? I thought you had at least surface and TOA? Why did you not show Q1-Q2 for the case studies where the terms are larger, and different from Fig 8? Typically Q1-Q2-QR has been interpreted as the upward transport of moist static energy, h, by moist convection, but you do not discuss this, nor the added complexity of convection within the diurnal cycle.

L314 time-lag... The vertically pointing cloud data are 'point' measurement (L148)? What is the effective spatial resolution of the omega field? Your discussion (L93 on) of spatial field analysis is vague, and gives no sense of the effective spatial and temporal resolution; and how the fields were effectively smoothed to get omega and other terms.

Alan Betts

---

## Referee Comment (RC2) · E. Souza (Referee) · 26 Aug 2016

The authors study the behavior of some convective fields during the GoAmazon experiment, focusing on the contrast between a rainy period and a dry period. Besides, a discussion on the diurnal cycle for some cases is presented. The article worth publication after some revisions.

In line 95, the authors mention the ECMWF analysis but no word about the horizontal

resolution. If the resolution is good enough, kinectic energy plays a non-neglectable role in the energy equation, and Q1 should be expressed in terms of potential temperature instead of static energy (see formulation in Yanai and Tomita, 1998, J. of Climate, p. 463).

Another important point is the radiative heating Qrad. In studies with less time resolution, an average value can be used. One can also argue that Qrad is small compared to Q1 and Q2. However, since this study addresses both the diurnal cycle and the vertical structure of the convective heating, and Qrad do undergoes a strong diurnal cycle and presents a vertical structure that impacts on the intensity of convection, the diurnal cycle of Qrad should be properly taken into account. That variable can be easily obtained from any numerical model.

The authors use domain-mean precipitation instead of point precipitation. In my opinion that is an outstanding advance of this study, since it provides a good framework to comparison with numerical model results for the region. Returning to the discusion of the previous paragraph, the vertical integral of Q1-Qrad, divided by the latent heat of evaporation, gives an estimative of the precipitation rate (see Eqn. 12a, by Yanai et al., 1973). This information could be easily obtained, and a comparison with the observed precipitation rate (investigating both the intensity and the correlation) could be performed.

In line 215, the authors point latent cooling due to ice melting as responsible for the minimum of Q1 observed around 600 hPa. What do the authors have to say about cumulus congestus whose top are around that level? That is a region of re-evaporation of water droplets and strong radiative flux divergence.
* * *

---

## Referee Comment (RC3) · J. Cohen (Referee) · 13 Sep 2016

The publication of this paper is relevant to the understanding of the diurnal cycle of convection in the Amazon region. However it is necessary that some doubtful points listed below are answered in order to accept the publication of this article.

Line 62: Give some reference about this paragraph.

Line 202: Greco et al, 1994 also estimated W, Q1 and Q2 for a Squall Line observed
during ABLE-2b in the Manaus region. I recommend the authors include this pioneering reference in the Amazon. Compare your results with those found in Greco et al, 1994.

Line 273: Include the paper Greco et al, 1990 which was the first paper that characterized the LOS, BOS and COS in Manaus region (Greco et al 1990).

Line 291: One of the case studies in this paper was called COS as defined by Kouky, 1980; Greco et al, 1990; Cohen et al, 1995. Thus, to be considered as a COS, this convective system must have formed in the afternoon of March 19 along the Atlantic Coast Amazon and crossed by Manaus on March 20 in the morning. Looking at satellite images I realized that actually formed a squall line on the coast in March 19, but during its propagation inland this convective system lost its linear format and did not reach Manaus region. Therefore, I recommend that the convective system observed March 20 be classified as BOS, since even in the satellite images it does not represent a COS classic. Thus, I ask you to choose one of two cases of BOS to use in this publication.

---

## Author Comment (AC1) · 17 Oct 2016

We would like to thank reviewers for helpful comments and suggestions. All comments and suggestions have been considered. The point-by-point responses are listed below:
* * *
Reply to the Interactive comments by Alan. K. Betts on 2 August, 2016

This is an interesting discussion of two IOPs during Go-Amazon, one in the wet season and the other the dry season; with three cases studies from the wet season of local, coastal and basin scale convection. It is suitable for publication with a little revision. It is written for a 'club' and needs clarity for less specialized readers.

**** You mention 2014-2015, but the only data you show is from 2014.

Reply: The GoAmazon2014/5 campaign is a 2-year experiment but the two IOPs that we studied were both conducted in the year 2014. This has been clarified in the revised text (lines 20-21).

**** Figure 1 has a 'potential site'? L93 - see below.

Reply: We have removed the unnecessary sites in the revised Figure1.

**** P10. Although equations (1) and (2) are based on historic literature, the units are not well defined here, and Q1, Q2 do not actually have the units of K/hr as in Figure 8. Nor does QR in (1).

Reply: Revised the equations (1), (2) and (3) (see revised equations below) to make the units consistent.

**** I did not find the Q1-Q2 discussion very satisfactory for Figure 8. Do you have an estimate for QR? I thought you had at least surface and TOA? Why did you not show Q1-Q2 for the case studies where the terms are larger, and different from Fig 8? Typically Q1-Q2-QR has been interpreted as the upward transport of moist static energy, h, by moist convection, but you do not discuss this, nor the added complexity of convection within the diurnal cycle.

Reply: As suggested, we revised Figure 8 by including Qrad and add relevant discussion there (line 299-307). In the paper, Qrad is estimated from using the radiative transfer model in the single-column model of CAM5 (Neale et al., 2012) driven by the large-scale forcing data derived from this study due to the lack of observations of Qrad.

**** L314 time-lag... The vertically pointing cloud data are 'point' measurement (L148)? What is the effective spatial resolution of the omega field? Your discussion (L93 on) of spatial field analysis is vague, and gives no sense of the effective spatial and temporal resolution; and how the fields were effectively smoothed to get omega and other terms.

Reply: This has been clarified in lines 136-142 and 161. The cloud data are "point" measurements while the omega field (and other large-scale fields) represents an average over the analysis domain, which is ∼110km in radius. The cloud data are measured at 67.8 km downwind of the center and ∼170 km downwind of the east edge of the domain. Therefore, it can only see a convective system after the system propagates into the domain and is captured by the field of the view of the radar.

————————————————————————————————-

Reply to the Interactive comments by Enio Souza on 26 August, 2016

The authors study the behavior of some convective fields during the GoAmazon experiment, focusing on the contrast between a rainy period and a dry period. Besides, a discussion on the diurnal cycle for some cases is presented. The article worth publication after some revisions.

**** In line 95, the authors mention the ECMWF analysis but no word about the horizontal resolution. If the resolution is good enough, kinetic energy plays a non-neglectable role in the energy equation, and Q1 should be expressed in terms of potential temperature instead of static energy (see formulation in Yanai and Tomita, 1998, J. of Climate, p. 463).

Reply: The ECMWF analysis we got is in 0.5° resolution but the energy budget we calculate is averaging over the GOAmazon domain which is ∼110km in radius (added in line 112-113). Giving that resolution, we believe that kinetic energy is neglectable in the energy equation and dry static energy is a good estimation to be used.

**** Another important point is the radiative heating Qrad. In studies with less time

resolution, an average value can be used. One can also argue that Qrad is small compared to Q1 and Q2. However, since this study addresses both the diurnal cycle and the vertical structure of the convective heating, and Qrad do undergoes a strong diurnal cycle and presents a vertical structure that impacts on the intensity of convection, the diurnal cycle of Qrad should be properly taken into account. That variable can be easily obtained from any numerical model.

Reply: Thanks for the suggestion. Following your suggestion, we have now estimated Qrad and accounted for it in the discussion of the diurnal cycle (line 271-277 and lines 299-307). In the revised paper, Qrad is estimated from using the radiative transfer model in the single-column model of CAM5 (Neale et al., 2012) driven by the large-scale forcing data derived from this study due to the lack of observations of Qrad.

**** The authors use domain-mean precipitation instead of point precipitation. In my opinion that is an outstanding advance of this study, since it provides a good framework to comparison with numerical model results for the region. Returning to the discussion of the previous paragraph, the vertical integral of Q1-Qrad, divided by the latent heat of evaporation, gives an estimative of the precipitation rate (see Eqn. 12a, by Yanai et al., 1973). This information could be easily obtained, and a comparison with the observed precipitation rate (investigating both the intensity and the correlation) could be performed.

Reply: Thank you for your comment. Because the thermodynamic equation and water vapor conservation equation are explicitly satisfied in the variational analysis and the observed precipitation is used as the constraint, the vertical integral of Q1-Qrad and vertical integral of Q2 are consistent with the observed precipitation rate implicitly. We have added a sentence to make this clear (lines 217-220).

**** In line 215, the authors point latent cooling due to ice melting as responsible for the minimum of Q1 observed around 600 hPa. What do the authors have to say about cumulus congestus whose top are around that level? That is a region of re-evaporation

of water droplets and strong radiative flux divergence.

Reply: Thanks. We agree that besides ice melting, there could be other reasons that cause the minimum of Q1 (or the double peak structure), such as the combining effect of lower level shallow cumulus and upper level deep convection or MCSs. The text has been revised by including these potential causes (lines 237-245): "For Q1, previous studies (Johnson, 1984; Schumacher et al., 2007) interpreted the double peaks as a result from shallow cumulus in lower level and deep convection or MCS in middle to upper level, although sometimes they superposed as one peak (Johnson, 1984). Moreover, latent cooling due to ice melting in the stratiform region may also contribute to the local minimum of Q1 which, in some field campaigns, is only shown as an inflection (Johnson et al., 2016). Nevertheless, the local minimum or the inflection usually occurs near the melting level (∼600 hPa) in many other tropical field campaigns (e.g. Schumacher et al., 2008; Xie et al., 2010a; Ahmed et al., 2016), indicating that the melting level is nearly constant in the tropics."

————————————————————————————————-

Reply to the Interactive comments by Julia Cohen on 13 September, 2016

The publication of this paper is relevant to the understanding of the diurnal cycle of convection in the Amazon region. However it is necessary that some doubtful points listed below are answered in order to accept the publication of this article.

**** Line 62: Give some reference about this paragraph.

Reply: Relevant references (Harriss et al., 1988; Harriss et al., 1990; Fu et al., 1999; Nobre et al., 2009; Filho et al., 2015) are added in the revised text (Lines 61-63)

**** Line 202: Greco et al, 1994 also estimated W, Q1 and Q2 for a Squall Line observed during ABLE-2b in the Manaus region. I recommend the authors include this pioneering reference in the Amazon. Compare your results with those found in Greco et al, 1994.

Reply: Thanks for the suggestion. We added this reference and compared the results in lines 223-227: "Overall, the magnitude of Q1 and Q2 are consistent with Schumacher et al. (2007) for LBA at southwestern Brazilian Amazon but much smaller than Greco et al. (1994) at Manaus region. The much larger magnitude in Greco et al. (1994) is likely because it is a case study of one day. The peak height in this study is also lower than the other two studies, indicating that our cases contain more shallow cumulus and convections with low-level heating and drying."

**** Line 273: Include the paper Greco et al, 1990 which was the first paper that characterized the LOS, BOS and COS in Manaus region (Greco et al 1990).

Reply: revised as suggested.

**** Line 291: One of the case studies in this paper was called COS as defined by Kouky, 1980; Greco et al, 1990; Cohen et al, 1995. Thus, to be considered as a COS, this convective system must have formed in the afternoon of March 19 along the Atlantic Coast Amazon and crossed by Manaus on March 20 in the morning. Looking at satellite images I realized that actually formed a squall line on the coast in March 19, but duringits propagation inland this convective system lost its linear format and did not reach Manaus region. Therefore, I recommend that the convective system observed March 20 be classified as BOS, since even in the satellite images it does not represent a COS classic. Thus, I ask you to choose one of two cases of BOS to use in this publication.

Reply: We re-examined the radar and satellite images and agree that the March 20 case is classified as BOS. Another COS case on Feb. 21 is chosen for analysis and the numbers of convective systems are also re-counted. Relevant table and figures are revised due to this change (Table 1, Figures 9 and 11). Our main results are not changed due to this revision.

[Figure]

$$Q_1 = \frac{1}{C_p}\left(\frac{\partial \overline{s}}{\partial t} + \overline{\overline{V}} \cdot \nabla \overline{s} + \overline{\omega}\frac{\partial \overline{s}}{\partial p}\right)$$

$$= \frac{1}{C_p}\left(Q_{rad} + L_v\left(c - e\right) - \frac{\partial \overline{\omega' s'}}{\partial p}\right) \tag{1}$$

$$Q_2 = -\frac{L_v}{C_p}\left(\frac{\partial \overline{q}}{\partial t} + \overline{\overline{V}} \cdot \nabla \overline{q} + \overline{\omega}\frac{\partial \overline{q}}{\partial p}\right)$$

$$= \frac{L_v}{C_p}\left(c - e + \frac{\partial \overline{\omega' q'}}{\partial p}\right) \tag{2}$$

$$Q_1 - Q_2 - Q_{rad} = -\frac{1}{C_p}\frac{\partial \overline{\omega' h'}}{\partial p} \tag{3}$$

Revised equations for unit consistency

[Figure]

Figure 1: The location of GoAmazon site in this study. The red octagon represents the analysis domain. Locations of observational sites are indicated by yellow pentagrams. Locations of cities are indicated by white dots.

**Fig. 2.** revised figure 1

[Figure]

Figure 9: SIPAM radar reflectivity snapshots (left) and time series of domain-mean precipitation (right) for three cases of precipitating systems. From top to bottom: LOS, COS and BOS. The black octagons indicate the GoAmazon domain, and the red arrows indicate the propagating direction of the system.

**Fig. 3.** revised figure 9
Interactive
comment

[Figure]

Figure 11: Similar as Figure 10 but for the COS case.

**Fig. 4.** revised figure 11

---

## Referee Report (RR1)

Large-Scale Vertical Velocity, Diabatic Heating and Drying Profiles Associated with Seasonal and Diurnal Variations of Convective Systems Observed in the GoAmazon2014/5 Experiment

Referee: Enio P. Souza

enio.souza@ufcg.edu.br

The manuscript has been improved upon its original version. I guess it good enough to be published in ACP, after some minor revision.

**Minor Points**

The authors should mention in the text the horizontal resolution of the ECMWF analysis data used in the study.

In Equation 1, 2 and 3, and thougout the text, use lower case for specific heat, instead of $C_p$, for it is more usual in literature.

Finally, a substantial grammar revision of the text is necessary.